# *Photosystem II Subunit S* overexpression increases the efficiency of water use in a field-grown crop

Katarzyna Głowacka[1,2], Johannes Kromdijk[1], Katherine Kucera[1], Jiayang Xie[1,3], Amanda P. Cavanagh[1], Lauriebeth Leonelli[4], Andrew D.B. Leakey[1,3,5], Donald R. Ort[1,6], Krishna K. Niyogi[4,7] & Stephen P. Long [1,8]

Insufficient water availability for crop production is a mounting barrier to achieving the 70% increase in food production that will be needed by 2050. One solution is to develop crops that require less water per unit mass of production. Water vapor transpires from leaves through stomata, which also facilitate the influx of $CO_2$ during photosynthetic assimilation. Here, we hypothesize that *Photosystem II Subunit S* (*PsbS*) expression affects a chloroplast-derived signal for stomatal opening in response to light, which can be used to improve water-use efficiency. Transgenic tobacco plants with a range of *PsbS* expression, from undetectable to 3.7 times wild-type are generated. Plants with increased *PsbS* expression show less stomatal opening in response to light, resulting in a 25% reduction in water loss per $CO_2$ assimilated under field conditions. Since the role of PsbS is universal across higher plants, this manipulation should be effective across all crops.

[1] Carl R. Woese Institute for Genomic Biology, University of Illinois, Urbana, IL 61801, USA. [2] Institute of Plant Genetics, Polish Academy of Sciences, 60-479 Poznań Poland. [3] Department of Crop Sciences, University of Illinois, Urbana, IL 61801, USA. [4] Howard Hughes Medical Institute, Department of Plant and Microbial Biology, University of California Berkeley, Berkeley, CA 94720, USA. [5] Department of Plant Biology, University of Illinois, Urbana, IL 61801, USA. [6] Photosynthesis Research Unit, US Department of Agriculture-Agricultural Research Service, University of Illinois, Urbana, IL 61801, USA. [7] Molecular Biophysics and Integrated Bioimaging Division, Lawrence Berkeley National Laboratory, Berkeley, CA 94720, USA. [8] Lancaster Environment Centre, University of Lancaster, Lancaster LA1 1YX, UK. These authors contributed equallly: Katarzyna Głowacka and Johannes Kromdijk. Correspondence and requests for materials should be addressed to K.K.N. (email: niyogi@berkeley.edu) or to S.P.L. (email: slong@illinois.edu)

Demand for primary foodstuffs, that is, grains and seeds of our major crops, may increase by 70–100% by 2050[1,2]. One major barrier to meeting this large demand will be availability of water for crop production. Crop productivity strongly depends on having a sufficient supply of freshwater, and agriculture consumes 90% of total global freshwater[3]. A large proportion of global food crops depend on irrigation, which is depleting global groundwater, and putting the sustainability of global food production at risk[4]. To capture atmospheric $CO_2$ during photosynthesis, stomatal pores need to stay open to allow $CO_2$ diffusion into the leaf. However, stomatal opening causes most of the water absorbed by plant roots to be lost via transpiration. Transpiration is proportional to the water vapor pressure deficit (VPD) from leaf to air, which represents the gradient between the humidity in leaf internal airspaces and drier air surrounding the leaf. With the global rise in air and surface temperatures, VPD has been increasing, thus increasing demand for irrigation[5,6].

Because stomatal opening controls both the $CO_2$ influx and the water vapor efflux, stomata have to respond to many different cues to balance the fluxes[7]. Progress has been made in unraveling the molecular basis of the response of stomata to intercellular $CO_2$ concentration[8] and blue light[9], but much less is known about stomatal response to light quantity. Stomatal opening in response to light is typically much less pronounced in detached epidermal layers, but can be restored when the connection with mesophyll cells is restored[10–12]. Therefore, although some control resides in the guard cells, stomatal responses to light intensity seem to rely strongly on a signal derived from the underlying mesophyll tissue. The rate of photosynthetic $CO_2$ assimilation at high light intensity is usually limited by the restriction of $CO_2$ influx imposed by stomata. Thus, control of stomatal opening by a signal derived directly from photosynthesis could provide a feedback loop to match the light energy processed by the photosynthetic light-dependent reactions with sufficient supply of $CO_2$. However, several mutants deficient in specific components of the photosynthetic light-dependent or carbon reactions typically show vast decreases in the rate of $CO_2$ assimilation without corresponding changes in stomatal conductance. For example, in tobacco plants containing reduced amounts of cytochrome $b_6/f$[13], ribulose-1,5-bisphosphate carboxylase-oxygenase (Rubisco)[13,14], glyceraldehyde 3-phosphate dehydrogenase[15], or sedoheptulose-bisphosphatase[16] net assimilation rate was substantially reduced, but stomatal conductance often remained relatively unaltered compared to the wild-type (WT). These results show clearly that the stomatal opening signal does not scale directly with the rate of $CO_2$ uptake. However, the interpretation of these results is severely complicated by the strong decrease of net $CO_2$ assimilation rate associated with these transgenic alterations, which greatly increases the $CO_2$ concentration in the intercellular airspaces within the leaf ($C_i$), providing a potent signal for stomatal closing[8].

A recent analysis suggested the redox state of chloroplastic quinone A ($Q_A$) as an early signal for stomatal opening in response to light[17], with a more reduced $Q_A$ pool corresponding to increased stomatal opening. $Q_A$ is the primary electron acceptor downstream of photosystem II and its oxidation state reflects the balance between excitation energy at photosystem II and the rate of the Calvin–Benson cycle. This predicts that decreasing the excitation pressure at photosystem II should directly affect stomatal opening in response to light by keeping $Q_A$ more oxidized. This prediction is tested here by altering expression of *Photosystem II Subunit S* (*PsbS*). *PsbS* expression directly affects the rate at which excitation energy absorbed by the antenna complex of photosystem II is used to reduce $Q_A$, because of its role in non-photochemical quenching (NPQ). NPQ protects

the photosynthetic machinery under excessive light conditions via controlled dissipation of absorbed light energy as heat[18,19]. *PsbS* expression strongly stimulates NPQ and promotes photoprotection under high light or rapidly fluctuating conditions[20], but typically does not affect steady-state rates of net $CO_2$ assimilation[21], thus keeping control of stomatal movements via $C_i$ relatively unaltered. Manipulation of *PsbS* expression thus provides an ideal test case to verify if $Q_A$ redox state is indeed an early signal for light-induced stomatal opening. Since PsbS stimulates the thermal dissipation of excitation energy, we predicted that increased expression of this protein would keep the redox state of $Q_A$ more oxidized, decrease stomatal opening in response to light, and decrease water loss at the leaf level. To test this hypothesis, *Nicotiana tabacum* lines with both increased and decreased *PsbS* expression were generated and analyzed under controlled and field conditions. *N. tabacum* cv "Petite Havana" was transformed with the coding sequence of *N. benthamiana PsbS* fused to the cauliflower mosaic virus 35S promoter for constitutive strong expression (Supplementary Fig. 1). Four independent, single-copy transformation events with increased NPQ amplitude (PSBS-28, PSBS-34, PSBS-43, and PSBS-46) were selected and selfed to obtain progeny homozygous for the transgene. Additionally, two events exhibiting spontaneous partial silencing of *PsbS* expression (psbs-4 and psbs-50) were selected for further analysis. Our results show that the light response of stomatal conductance is clearly affected by *PsbS* expression. Plants overexpressing *PsbS* show an average 25% reduction in water loss per $CO_2$ assimilated under field conditions.

## Results

**PsbS expression under controlled conditions**. PSBS-28, PSBS-43, psbs-4, and WT *N. tabacum* plants were grown in a controlled-environment cabinet and *PsbS* transcript and protein levels were measured in samples from the youngest fully expanded leaves. PSBS-28 and PSBS-43 samples showed 4.2-fold and 3.5-fold increases in total (transgenic and native) *PsbS* transcript relative to WT (Fig. 1a), whereas transcript levels in psbs-4 were 10-fold less than WT. PsbS protein expression, normalized to the large subunit of the oxygen-evolving complex (PsbO) as a relative measure for the abundance of photosystem II, was 2.7-fold higher in PSBS-43 and 3.5-fold higher in PSBS-28 relative to WT while virtually absent in psbs-4 (Fig. 1b, c).

**PsbS expression affects intrinsic water-use efficiency**. Net $CO_2$ uptake ($A_n$) increased in response to light intensity until approximately $800 \, \mu mol \, m^{-2} \, s^{-1}$ and was not significantly affected by *PsbS* expression ($P = 0.6$, analysis of variance (ANOVA); Fig. 1d). The maximum capacity for carboxylation of ribulose-bisphosphate ($V_{cmax}$) and the maximum rate of whole-chain electron transport ($J_{max}$) showed weak positive trends with PsbS content (Supplementary Fig. 2a–d). Rubisco content was similar between lines (Supplementary Fig. 2e), but Rubisco activation state was slightly lower in psbs-4 (85%), compared to WT (95%), whereas the overexpressing lines were similar to wild-type (PSBS-28, 93%) or slightly higher (PSBS-43, 102%; Supplementary Fig. 2f). Stomatal limitation to net $CO_2$ assimilation also significantly differed with PsbS content (Supplementary Fig. 2g); however, the aforementioned changes in photosynthetic capacity and Rubisco biochemistry counteracted these differences, leaving $A_n$ unchanged between all lines.

As expected, differences in PsbS protein expression led to pronounced differences in NPQ (Fig. 1e). At high light, NPQ was significantly higher in PSBS-28 and PSBS-43 relative to WT ($P \leq 0.05$, Dunnett's two-way test) and significantly lower in psbs-4 ($P \leq 0.02$, Dunnett's two-way test). In concert with these

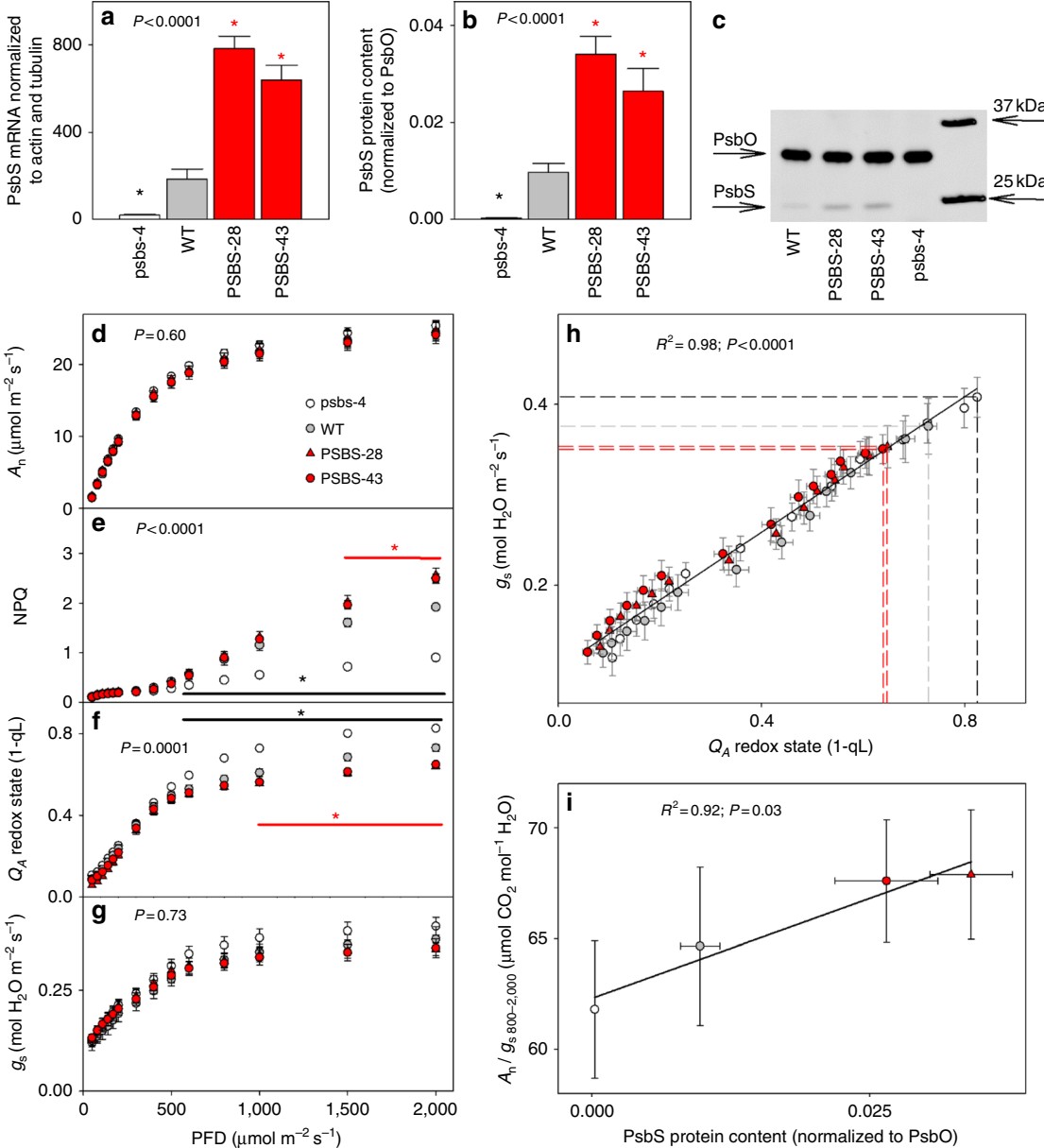

**Fig. 1** Photosynthesis and water-use efficiency in *Nicotiana tabacum* plants with modified *PsbS* levels. **a** *PsbS* mRNA levels normalized to actin and tubulin sampled from fully expanded leaves of psbs-4, PSBS-28, PSBS-43 and wild-type (WT) *N. tabacum* plants. **b** PsbS protein levels normalized to the large subunit of oxygen-evolving complex of photosystem II (PsbO), determined from densitometry on immunoblots. **c** Representative immunoblot for PsbS and PsbO. **d** Net $CO_2$ fixation rate ($A_n$), **e** NPQ, **f** quinone A ($Q_A$) redox state, and **g** stomatal conductance ($g_s$) as a function of incident light intensity in fully expanded leaves. **h** Linear correlation between $Q_A$ redox state and $g_s$. Broken lines indicate measurements at highest light intensity. **i** Linear correlation between PsbS protein levels and intrinsic water-use efficiency ($A_n/g_s$) at light intensity above $600 \, \mu mol \, m^{-2} \, s^{-1}$. Asterisks/lines show significant differences from WT (black for silencing, red for overexpressing lines; Dunnett's two-way test; $\alpha = 0.05$). Error bars indicate s.e.m. ($n$ = from 6 to 10 biological replicates)

differences, the redox state of $Q_A$ was significantly more oxidized in PSBS-28 and PSBS-43 relative to WT ($P \leq 0.02$, Dunnett's two-way test; Fig. 1f) and more reduced in psbs-4 ($P \leq 0.002$, Dunnett's two-way test; Fig. 1f). These differences in $Q_A$ redox state at high light were reflected in differences in stomatal conductance ($g_s$; Fig. 1g). The change in $g_s$ was consistent with altered regulation of stomatal opening rather than any changes in stomatal or epidermal anatomy, that is, pore dimensions or stomatal density. Stomatal density was 21% (abaxial) to 23% (adaxial) lower in psbs-4 and 18% (abaxial) lower in PSBS-43, relative to WT ($P = 0.006$ for abaxial and $P = 0.02$ for adaxial, ANOVA; Supplementary Fig. 3a), but unchanged in PSBS-28.

Stomatal pore dimensions on both abaxial and adaxial leaf surfaces were very similar between all lines (Supplementary Fig. 3b–d). In addition, all measurements of $Q_A$ redox state and $g_s$ in all lines could be described by a single highly significant positive correlation ($P < 0.0001$, ANOVA; Fig. 1h), consistent with a role for $Q_A$ redox state as an early determinant for stomatal opening in response to light intensity[17]. Furthermore, the differences in $g_s$ associated with PsbS expression in combination with unchanged $A_n$ resulted in a strong correlation between intrinsic water-use efficiency ($A_n/g_s$, WUEi) and PsbS expression ($R^2 = 0.92$, $P = 0.03$, ANOVA; Fig. 1i).

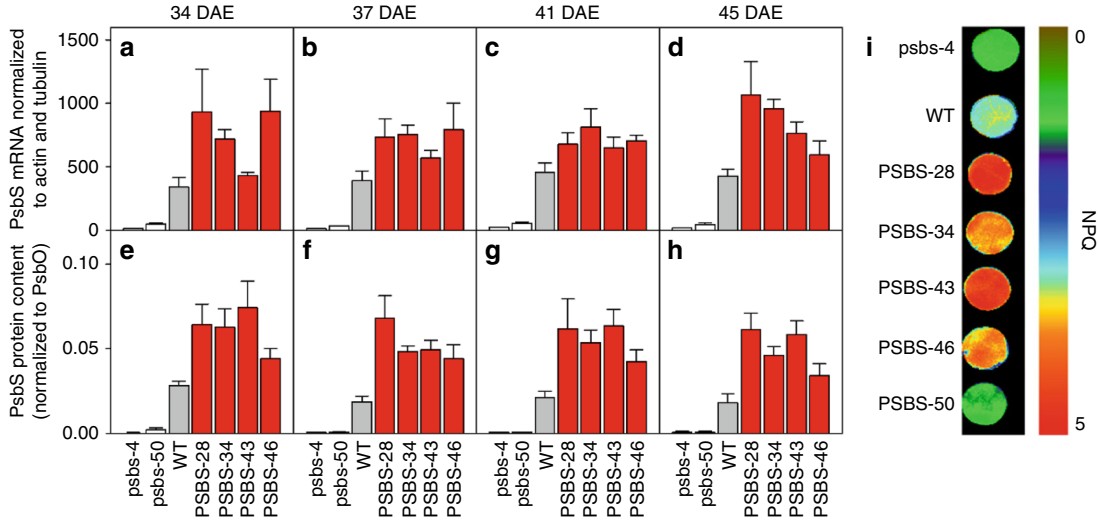

**Fig. 2** *PsbS* expression and photoprotection in *Nicotiana tabacum* plants grown under field conditions. **a–d** *PsbS* mRNA levels and **e–h** PsbS protein levels in several tobacco genotypes with modified *PsbS* expression levels as well as wild-type (WT) tobacco at four time points during the field experiment (DAE = days after emergence). **i** Representative levels of NPQ for each genotype determined on leaf discs. All genotype means were significantly different from WT (Dunnett's two-way test; $P \leq 0.008$ for mRNA; $P \leq 0.001$ for protein). Error bars indicate s.e.m. ($n = 4$ biological replicates)

**WUE and productivity under field conditions.** The differences in WUEi observed under controlled conditions were subsequently tested under field conditions. A field experiment was conducted with transformants with both increased (PSBS-28, PSBS-34, PSBS-43, and PSBS-46) and decreased *PsbS* expression (psbs-4 and psbs-50) in an incomplete block design (Supplementary Fig. 4). Western blotting confirmed differences in PsbS expression in the youngest fully expanded leaf for each genotype at 34, 37, 41, and 45 days after emergence (DAE) (Fig. 2a–h). As predicted, transformants showed a broad range of *PsbS* expression from almost none to 3.7-fold higher than WT, which were directly reflected in levels of NPQ measured on leaf discs (Fig. 2i). Critically, as under controlled conditions, net $CO_2$ assimilation rate did not differ among the transformants and WT (Fig. 3a, Supplementary Fig. 5a), but $g_s$ correlated negatively to *PsbS* expression ($P = 0.0001$, ANOVA; Fig. 3b and Supplementary Fig. 5b). The reduction in $g_s$ due to PsbS overexpression varied between 4 and 30%, whereas $g_s$ was increased by 46% due to decreased PsbS expression (Fig. 3b). Once again, WUEi was significantly affected by genotype ($P = 0.007$, ANOVA; Fig. 3c and Supplementary Fig. 5d) and correlated positively with PsbS expression ($R^2 = 0.94$, $P = 0.004$, ANOVA; Fig. 3d). Increased PsbS expression resulted in 25–33% increased WUEi, whereas decreased *PsbS* expression led to 14% reduction in WUEi. Final size and dry weight were determined at the end of the field experiment. All biomass productivity traits were significantly affected by *PsbS* expression ($P \leq 0.008$, ANOVA; Fig. 3e–g). Decreased *PsbS* expression significantly reduced dry weight (22%, $P \leq 0.002$, Dunnett's two-way test; Fig. 3e, Supplementary Fig. 6), leaf area (15%, significant only in psbs-50, $P = 0.03$, Dunnett's two-way test; Fig. 3f) and plant height (15%, $P < 0.0001$, Dunnett's two-way test; Fig. 3g). The productivity measures in transformants with increased *PsbS* expression did not show a consistent response. PSBS-28 showed significant decreases in dry weight ($-18\%$, $P = 0.008$, Dunnett's two-way test; Fig. 3e), leaf area ($-19\%$, $P = 0.005$, Dunnett's two-way test; Fig. 3f), and plant height was also significantly smaller in PSBS-28 and PSBS-34 ($-9$ and $-8\%$, $P \leq 0.01$, Dunnett's two-way test; Fig. 3g), whereas the same productivity measures were not significantly affected in PSBS-43 and PSBS-46, relative to WT.

## Discussion

Our results provide direct proof, through genetic manipulation, that increasing *PsbS* expression suppresses stomatal opening with little effect on $CO_2$ uptake and so increases WUE. We showed a strong dependence of $g_s$ on *PsbS* expression (Figs. 1g, 3b) and that overexpression of *PsbS* significantly improved WUEi (Fig. 3c), representing a strong decrease (averaging 25%) in the amount of water used for each molecule of $CO_2$ assimilated at leaf level by an irrigated field crop. Novel bioengineering strategies to improve crop WUE such as exemplified here are urgently needed, especially considering the long timelines for developing new crop varieties[22]. Although this test of concept was performed on tobacco, the role of PsbS in NPQ is universal across higher plants[23], so this manipulation can be expected to be effective across all crops.

Here a large improvement of leaf level WUEi is shown (Figs. 1i, 3c), which can be expected to conserve soil moisture and may result in increased productivity if the crop becomes water-limited. However, many feedbacks could lessen this improvement at the whole crop level. Open-air elevation of $CO_2$ has provided a direct test of the significance of such feedbacks. When stomatal conductance in a mature soybean canopy was reduced by 10% due to open-air elevation of $[CO_2]$, this led to a corresponding decrease in the measured ecosystem evapotranspiration by 8.6%[24].

Our data for the first time show that modulation of $Q_A$ redox state via *PsbS* expression affects stomatal conductance and leaf WUE. Interestingly, manipulation of *PsbS* expression did not directly impact steady-state photosynthesis, which is consistent with previous findings[21] and thus allowed for specific targeting of the putative $Q_A$-redox signal without strongly affecting the $CO_2$ control over stomatal conductance via $C_i$ (Fig. 4a). Although $C_i$ was still somewhat affected (Supplementary Fig. 5c), there was a strong linear correlation of $Q_A$ redox state with $g_s$ (Fig. 1h), in contrast to several previous manipulations, where blockage of electron transfer components downstream of $Q_A$[13,25] or impairment of Calvin–Benson cycle function[14–16,26,27] strongly affected stomatal control by both $Q_A$ redox state and $C_i$ in opposing ways, resulting in ambiguous effects on stomatal conductance. Furthermore, when electron transfer is blocked upstream of $Q_A$, both $C_i$ and $Q_A$-redox signals stimulate stomatal closure, which matches observations of stomatal conductance in plants with reduced

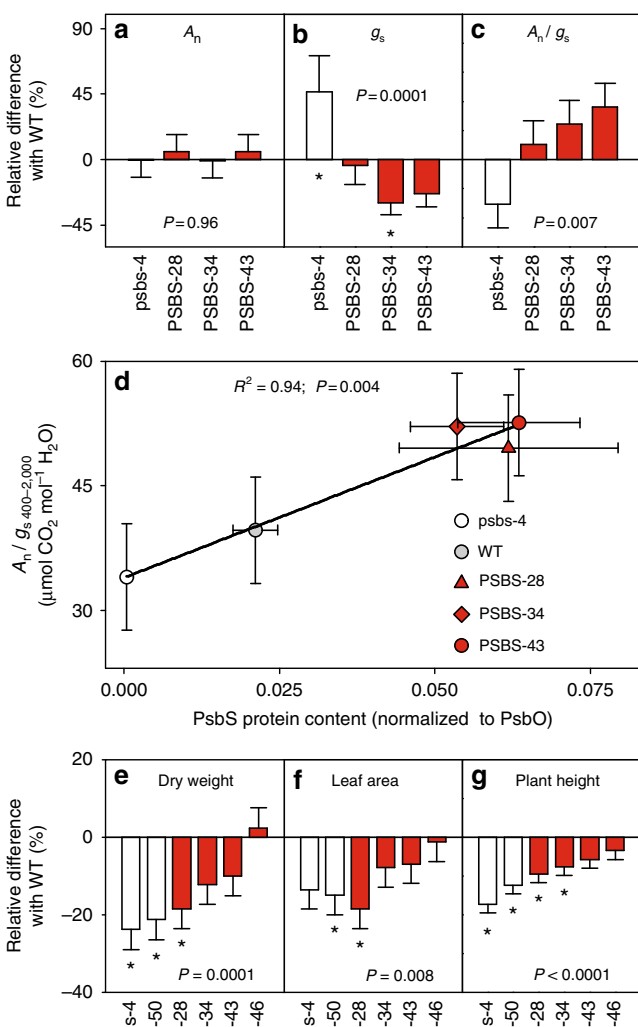

**Fig. 3** Photosynthetic water-use efficiency and productivity of field-grown *Nicotiana tabacum* plants with modified PsbS levels. **a** Net $CO_2$ fixation rate ($A_n$), **b** stomatal conductance ($g_s$), **c** intrinsic water-use efficiency ($A_n/g_s$) in tobacco genotypes with modified *PsbS* expression levels relative to wild-type tobacco. **d** Linear correlation between PsbS protein levels and $A_n/g_s$ at light intensity above 300 $\mu$mol m$^{-2}$ s$^{-1}$. **e** Total dry weight, **f** Leaf area, and **g** plant height. Error bars indicate s.e.m. (**a–d** $N = 4$ biological replicates; **e–g** $n = 6$ blocks for transgenic and $n = 12$ blocks for WT), and asterisks indicate significant differences between transgenic lines and WT (Dunnett's two-way test; $\alpha = 0.05$), $P$-values indicate significance of line effect in ANOVA (**a–c** and **e–g**) or significance of regression (**d**)

levels of PsbO[28] or treated with DCMU (3-(3,4-dichlorophenyl)-1,1-dimethylurea)[29]. Thus, when the stomatal control by $Q_A$ redox and $C_i$ are both accounted for, as shown by the schematic in Fig. 4b, our results complement previous observations and for the first time show experimental evidence for molecular regulation of stomatal response to light intensity. Although a constitutive promoter was used here, and thus guard cell expression of PsbS was also affected, it seems most likely that the $Q_A$-redox signal to open stomata in response to light is perceived in the mesophyll, especially considering that the fluorescence signal used to estimate $Q_A$-redox state is primarily originating from the chloroplast-rich mesophyll tissue, consistent with several previous investigations[7].

Since water supply was not limiting for growth in our field experiment, the increased WUE associated with *PsbS*

overexpression was not advantageous for biomass productivity, but instead led to slight decreases in final plant size and dry weight (Fig. 3e–g and Supplementary Fig. 6). We hypothesize that the increase in PsbS and associated increased levels of NPQ may have adversely affected the light-use efficiency of $CO_2$ assimilation under fluctuating light as previously shown in rice[30]. We have previously demonstrated that this efficiency is an important determinant of biomass productivity of tobacco under similar field conditions[31] and therefore may explain these findings.

Since $g_s$ directly affects the supply of $CO_2$ to photosynthesis, decreases in $g_s$ often result in decreased $A_n$[32]. Interestingly, the effects of PsbS on stomatal conductance did not translate into differences in $A_n$, even though $CO_2$ supply to photosynthesis was slightly affected (Supplementary Fig. 2g). Instead, maximum RuBP carboxylation capacity ($V_{cmax}$) and the maximal rate of linear electron transport ($J_{max}$) showed weak positive relationships with the amount of PsbS (Supplementary Fig. 2c, d), consistent with previous findings in rice with altered PsbS levels[30] and Rubisco activation state also showed a weak positive trend with PsbS content (Supplementary Fig. 2f). These results indicate that plants may be able to compensate for the effects of a decrease in stomatal conductance on $CO_2$ uptake by increasing photosynthetic capacity, thereby limiting the negative feedback on biomass productivity.

## Methods

**Plant material**. WT *N. tabacum* cv. "Petit Havana" seeds carrying TMV resistance (NN) were a gift from Professor Spencer Whitney. Lines exhibiting increased or reduced expression of PsbS were generated within this study as described below.

**Recombinant DNA and transformation**. The *N. benthamiana* PsbS gene coding sequence (www.uniprot.org, Q2LAH0_NICBE) was cloned in between the cauliflower mosaic virus 35S and octopine synthase terminator in the pEARLYGATE 100 binary vector. The resulting binary vector pEG100-NbPsbS conferred microbial resistance to kanamycin and bialaphos resistance *in planta* (Supplementary Fig. 1). *Nicotiana tabacum* cv. "Petite Havana" was transformed with pEG100-NbPsbS using the *Agrobacterium tumefaciens*-mediated protocol[33]. Copy number and homozygosity were assessed using digital droplet PCR[34]. Results shown are for homozygous offspring unless otherwise described.

**Transcription and protein expression**. Five leaf discs (total 2.9 cm$^2$) were from the youngest fully expanded leaf of five plants per genotype (controlled conditions) or four plants per genotype (field). Samples were taken 2 h after the start of the photoperiod. Protein and mRNA were extracted from the same leaf sample (NucleoSpin RNA/Protein kit, REF740933, Macherey-Nagel GmbH & Co., Düren, Germany). Extracted mRNA was treated by DNase (Turbo DNA-free kit; AM1907, Thermo Fisher Scientific, Waltham, MA, USA) and transcribed to cDNA using Superscript III First-Strand Synthesis System for RT-PCR (18089-051, Thermo Fisher Scientific, Waltham, MA, USA). Quantitative reverse transcription PCR was used to quantify *PsbS* transcripts (5′-GGCACAGCTGAATCTTGAAAC-3′ and 5′-CAGGGACAGGGTCATCAATAAA-3′) relative to *NtActin* (5′-CCTCACA-GAAGCTCCTCTTAATC-3′ and 5′-ACAGCCTGAATGGCGATATAC-3′) and *NtTubulin* (5′-GTACATGGCCTGTTGTTTGATG-3′ and 5′-CTGGATGGTCCTCTTTGTCTTT-3′).

Total protein concentration was quantified using a protein quantification assay (ref. 740967.50, Macherey-Nagel GmbH & Co., Düren, Germany). Samples containing 1 µg total protein were separated by sodium dodecyl sulfate-polyacrylamide gel electrophoresis electrophoresis, blotted to membrane (Immobilon-P, IPVH00010, Millipore, Tullagreen, Carrigtwohill, Ireland) using semi-dry blotting (Trans-Blot SD, Bio-Rad, Hercules, CA, USA), and sequentially immuno-labeled with primary antibodies raised against *At*PsbS (1:2,000 dilution; AS09533, Agrisera, Vännäs, Sweden) and *At*PsbO (1:20,000 dilution; AS06142-33, Agrisera, Vännäs, Sweden) followed by incubation with secondary antibodies (1:2,500 dilution; W401B, Promega, Madison, WI, USA). The sequential use of the two primary antibodies was verified empirically against blots where only one antibody was used and dilution series were used to establish the quantifiable range. Chemiluminescence was detected using a scanner (ImageQuant LAS-4010, GE Healthcare Life Sciences, Pittsburgh, PA, USA). A protein ladder (Precision Plus Protein Kaleidoscope Prestained Protein Standards, #1610375, Bio-Rad, Hercules, CA, USA) was used as a size indicator on each gel. Protein bands were quantified using densitometry with ImageQuant TL software (version 7.0 GE Healthcare Life Sciences, Pittsburgh, PA, USA) (Data Set 2 and 11 in Data Repository; https://data.mendeley.com/datasets/nsbjps9rkg/draft?a = 10508d31-685a-4a62-8f96-cb591c569e97). PsbS expression was normalized based on PsbO bands.

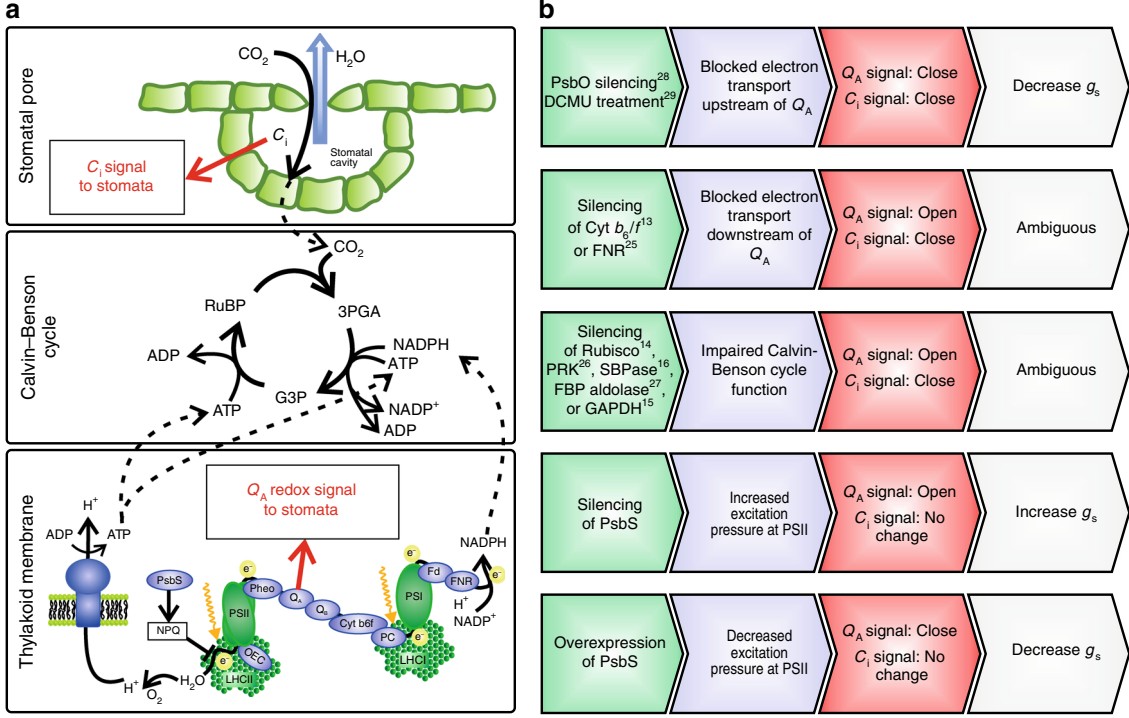

**Fig. 4** Interactions between light and $CO_2$ control over stomatal movements. **a** Schematic representation of processes in the chloroplast thylakoid membrane, Calvin–Benson cycle in the chloroplast stroma, and at the interchange between intercellular airspace and atmosphere through the stomatal pore. Indicated are where the $CO_2$ signal ($C_i$ signal) and the proposed $Q_A$-redox signal originate. **b** Chain of events showing the direct effects on the stomatal control signals from $C_i$ and $Q_A$ redox state of the manipulation of *PsbS* expression in the current work and several previously published manipulations, and the subsequent direction of change in stomatal conductance ($g_s$). Superscripted numbers indicate corresponding literature references. Abbreviations: 3PGA – 3-Phosphoglycerate; ADP – adenosine diphosphate; ATP – adenosine triphosphate; Cyt $b_6f$ – Cytochrome $b_6f$; DCMU – (3-(3,4-dichlorophenyl)-1,1-dimethylurea); FBP aldolase – Fructose-bisphosphate aldolase; Fd – Ferredoxin; FNR – Ferredoxin NADP(+) reductase; G3P – Glyceraldehyde-3-phosphate; GAPDH – Glyceraldehyde 3-phosphate dehydrogenase; LHCI or II – light-harvesting complex I or II; NADP+ / NADPH – nicotinamide adenine dinucleotide phosphate (oxidized/reduced); NPQ – non-photochemical quenching; OEC – Oxygen evolving complex; PC – Plastocyanin; Pheo – pheophytin; PRK – Phosphoribulokinase; PsbO – subunit of the oxygen evolving complex (OEC); PsbS – Photosystem II subunit S; PSI or II – photosystem I or II; $Q_A$ – Plastoquinone A; $Q_B$ – Plastoquinone B; Rubisco – Ribulose-1,5-bisphosphate carboxylase-oxygenase; RuBP – Ribulose 1,5-bisphosphate, SBPase – Sedoheptulose-bisphosphatase

**Photosynthetic gas exchange under controlled conditions.** Seedlings of psbs-4, PSBS-43, PSBS-28, and WT were germinated on growing medium (LC1 Sunshine mix, Sun Gro Horticulture, Agawam, MA, USA) in a controlled-environment walk-in growing chamber (Environmental Growth Chambers, Chagrin Falls, OH, USA) with photoperiod set to 12 h and temperature controlled at 23/18 °C (day/night). Five days after germination, psbs-4 seedlings with low NPQ were identified through chlorophyll fluorescence imaging and together with PSBS-28, PSBS-43, and WT seedlings transplanted to 3.8-L pots and randomly positioned in a controlled-environment chamber (PGC20, Conviron, Winnipeg, MB, Canada) with photoperiod set to 16 h and air temperature controlled at 20/25 °C (night/day). Light intensity at leaf-level was controlled at 500 μmol m$^{-2}$ s$^{-1}$. Plants were watered and plant positions were repositioned at random every 2 days until the fifth leaf was fully expanded. Gas exchange measurements were performed using an open gas exchange system (LI6400XT, LI-COR, Lincoln, NE, USA) equipped with a 2-cm$^2$ leaf chamber and integrated modulated fluorometer. All chlorophyll fluorescence measurements were performed using the multiphase flash routine[35]. To determine the light response of $A_n$ and whole-chain photosynthetic electron transport, gas exchange and pulse amplitude-modulated chlorophyll fluorescence were measured at a range of light intensities. Block temperature was controlled at 25 °C, [$CO_2$] inside the cuvette was maintained at 380 μmol mol$^{-1}$ and leaf-to-air water VPD was controlled to 1.1–1.4 kPa. Leaves were clamped in the leaf cuvette and dark-adapted for 1 h, after which minimal ($F_o$) and maximal fluorescence ($F_m$) were measured to determine maximal efficiency of whole-chain electron transport[36] ($F_v/F_m$, Eq. 1)

$$F_v/F_m \; = \; (F_m - F_o)/F_m . \tag{1}$$

Subsequently, light intensity (100% red LEDs, $\lambda_{peak}$ 630 nm) was slowly increased from 0 to 50, 80, 110, 140, 170, 200, 300, 400, 500, 600, 800, 1,000, 1,500, and 2,000 μmol m$^{-2}$ s$^{-1}$. When steady state was reached, $A_n$, $g_s$, and $C_i$ were logged, and $F'$ and $F_m'$ were measured to estimate the operating efficiency of

whole-chain electron transport[36] ($F_q'/F_m'$, Eq. 2). Since stomatal movements can include very long-term diurnal components[37,38], our routine was aimed at measuring only relatively short-term stomatal responses to changes in light intensity, and steady-state waiting times were kept between 10 and 20 min per step. NPQ of chlorophyll fluorescence was determined according to Eq. 3 assuming a Stern–Volmer quenching model[39]. Minimal fluorescence without dark adaptation ($F_o'$) was also determined (using a short far-red pulse to fully oxidize $Q_A$). The fluorescence parameter qL (Eq. 4) was used to estimate the fraction of $Q_A$ in its oxidized state (and correspondingly, $Q_A$ redox state as 1 – qL). The derivation of this parameter is assuming a "lake" model for photosynthetic antenna complexes (i.e., antennae are shared between reaction centers)[40]:

$$F_q'/F_m' \; = \; (F_m' - F')/F_m' , \tag{2}$$

$$NPQ \; = \; F_m/F_m' - 1, \tag{3}$$

$$q_L \; = \; (1/F' - 1/F_m')/(1/F_o' - 1/F_m') . \tag{4}$$

To evaluate the $CO_2$ response of $A_n$, leaves were allowed to reach steady state at a light intensity of 2,000 μmol m$^{-2}$ s$^{-1}$ (100% red LEDs, $\lambda_{peak}$ = 630 nm), with block temperature controlled at 25 °C and [$CO_2$] in the airstream set to 400 μmol mol$^{-1}$. Subsequently, [$CO_2$] was varied from 400 to 300, 200, 100, 75, 400, 400, 500, 600, 700, 800, 1,000, 1,200, and 1,500 μmol mol$^{-1}$. When steady state was attained, $A_n$, $g_s$, and $C_i$ were logged. $V_{cmax}$ was determined from the response of $A_n$ to chloroplastic $CO_2$ concentration ($C_c$) by fitting a biochemical model[41] with temperature corrections[42] to measurements. $C_c$ required an estimate of mesophyll conductance to $CO_2$ transfer ($g_m$). This was estimated independently for each point in the $CO_2$ response curve from parallel chlorophyll fluorescence measurements according to the variable $J$ method[43]. $J_{max}$ was determined by fitting a non-rectangular hyperbola to light response curves of linear electron transport estimated from chlorophyll fluorescence[44]. Stomatal limitation of $A_n$ was computed

using measurements at ambient $CO_2$ ($C_a = 380$ µmol mol$^{-1}$) and saturating light intensity, and predicted values of $A_n$ when stomata are not limiting (i.e., $C_i$ would equal $C_a$)[44].

**Rubisco activation state and content**. Plants were grown under controlled conditions as described above. Youngest fully expanded leaves were clamped in the cuvette of an open gas exchange system (LI6400XT with 2 × 3 LED light source), with light intensity set to 1800 µmol m$^{-2}$ s$^{-1}$, $CO_2$ concentration set to 400 µmol mol$^{-1}$, and block temperature set to 25 °C. After steady-state gas exchange was reached, leaves were rapidly removed and a disc of 0.55 cm$^2$ from the center of the portion of the leaf that had been enclosed in the cuvette was snap frozen in liquid N. Rubisco activity was determined by the incorporation of $^{14}CO_2$ into acid-stable products at 25 °C following an existing protocol[45]. Samples were ground in ten-broek glass homogenizers with ~2 mL cm$^{-2}$ $CO_2$-free extraction buffer containing 100 mM Hepes-KOH (pH 7.5), 2 mM Na$_2$ethylenediaminetetraacetic acid (EDTA), 20 mM MgCl$_2$, 5 mM dithiothreitol (DTT), 5 mg mL$^{-1}$ polyvinyl pyrrolidine, 15 mM amino-$n$-caproic acid and 3.5 mM benzamidine, and 5% v/v protease inhibitor cocktail (P9599, Sigma, St. Louis, MO, USA). Within 30 s of extraction, samples were assayed for initial Rubisco activity in a buffer containing 100 mM Bicine-NaOH (pH 8.2), 1 mM Na$_2$EDTA, 20 mM MgCl$_2$, 5 mM DTT, 1 mM ATP, 0.5 mM ribulose-1,5-bisphosphate, and 12.8 mM NaH$^{14}CO_3$ (15 Bq nmol$^{-1}$, Vitrax, Placentia, CA, USA). Assays were run for 30 s and terminated with the addition of 300 µL 5 N formic acid. The radioactivity of acid-stable products was determined by liquid scintillation counting (Packard Tri-Carb 1900 TR, Canberra Packard Instruments Co., Downers Grove, IL, USA). After determining initial activity, the extract was incubated with 10 mM NaHCO$_3$ and 20 mM MgCl$_2$ for 20 min at room temperature, and the total activity of the extract was assayed as above. Unless stated otherwise, all other reagents were purchased from Sigma (St. Louis, MO, USA). Purified RuBP was used in both initial and total activity assays to avoid under-estimating the activation state[46]. The activation state of Rubisco is determined by the ratio of initial to total activity. Rubisco content was determined from carba-mylated samples extracted as above using a [$^{14}$C]carboxy-arabinitol bisphosphate-binding assay[47] with a specific activity of 583 Bq nmol$^{-1}$ Rubisco, assuming eight binding sites per Rubisco[45].

**Stomatal density and stomatal complex dimension**. Plants were grown under controlled conditions as described above. Fresh leaf samples were taken from the youngest fully expanded leaf and mounted onto a microscope slide using double-sided tape. Topographies of the adaxial and abaxial surfaces were measured using a µsurf explorer optical topometer (Nanofocus, Oberhausen, Germany). The 20×/0.60 objective lens (image size 0.8 × 0.8 mm$^2$) and the 50×/0.80 objective lens (image size 0.32 × 0.32 mm$^2$) were used, respectively, for stomatal density quantification and measurements of stomatal complex dimensions. Based on a prior bootstrap analysis, 8 and 10 images were analyzed for each of four biological replicates for stomatal density quantification and measurements of stomatal complex dimensions, respectively. For stomatal density quantification, raw images were first optimized for light contrast in µsoft analysis software (Nanofocus, Oberhausen, Germany) and then exported into TIF format. Each stoma was labeled and counted manually using multi-point function in ImageJ (ImageJ 1.51k, NIH, Rockville, MD, USA). Stomatal density was derived by dividing the number of stomata in each image by the image size (0.64 mm$^2$). To measure the length and width of the stomatal complex, the distance measurement function in µsoft analysis software was applied on each stoma which could be completely observed in the image. Lines were drawn manually from end to end of the stomatal complex ellipse to measure stomatal length and width.

**Seedling propagation for field experiment**. Homozygous T$_2$ (psbs-50 and PSBS-28) or T$_3$ (PSBS-34, PSBS-43, and PSBS-46) seeds as well as segregating T$_1$ seed from psbs-4 and WT seed from the same harvest date were sown in the greenhouse on May 16, 2016. Five days after germination, seedlings exhibiting severely reduced NPQ were identified by chlorophyll fluorescence imaging of the psbs-4 T$_1$ progeny. These low NPQ seedlings as well as seedlings from all other lines and WT were propagated hydroponically for 2 weeks in floating trays (Transplant Tray GP009 6 × 12 cells, Speedling Inc., Ruskin, FL, USA) filled with specialized growing medium for hydroponics (Pro-mix PGX, Premier Tech, Quakertown, PA, USA). The concentration of total dissolved solids in the solution was measured every 2 days with a handheld TDS meter (COM-100, HM Digital Inc., Culver City, CA, USA) and adjusted to 100 ppm by the addition of 20-10-20 water-soluble fertilizer (Jack's Professional, JR Peters Inc., Allentown, PA, USA). Five days after the transplant to trays, Etridiazole fungicide (Terramaster 4EC to a final concentration of 78 µl L$^{-1}$, Crompton Manufacturing Company Inc., Middlebury, CT, USA) was added to the solution to protect the plants against root fungus disease in the field. Two applications of Mancozeb (Dithane Rainshield Fungicide at 1 g L$^{-1}$, Dow AgroSciences Canada Inc., Calgary, AB, Canada) were applied 6 and 9 days after transplant to prevent foliar fungus disease. On the same days, seedlings were sprayed with fermentation solids and solubles from *Bacillus thuringiensis*, subsp *israelensis*, strain AM65-52 (Gnatrol WDG Biological larvicide at 1 mL L$^{-1}$, Valent Biosciences Corp., Libertyville, IL, USA) to reduce the greenhouse population of fungus gnats.

**Field experimental design**. Seedlings were transplanted to an experimental field site at the University of Illinois Energy Farm (40.11°N, 88.21°W) on June 9, 2016. The field was prepared 2 weeks prior to transplant by rototilling, cultivation, and harrowing. At this time, chlorpyrifos (1.5 g m$^{-2}$ Lorsban 15 G Insecticide, Dow AgroSciences Canada Inc., Calgary, AB, Canada) was worked into the soil to suppress cutworm damage, sulfentrazone (29 µL m$^{-2}$ Spartan 4 F preemergence herbicide, FMC Agricultural Solutions, Philadelphia, PA, USA) was applied to reduce the emergence of weeds and slow-release fertilizer (30.8 g m$^{-2}$ ESN Smart Nitrogen, Agrium US Inc., Denver, CO, USA) was put down. After transplant, all seedlings were sprayed with thiamethoxam (7 mg/plant Platinum 75 SG insecticide, Syngenta Crop Protection LLC, Greensboro, NC, USA) to prevent damage from insect herbivory, and 12 days after the field transplant, all plants were sprayed with fermentation solids, spores, and insecticidal toxins from *Bacillus thuringiensis*, subsp. *kurstaki*, strain ABTS-351 (2.6 mL L$^{-1}$ DiPel Pro dry flowable biological insecticide, Valent Biosciences Corp.) to suppress tobacco hornworm. The field experiment was set up as an incomplete randomized block design with 12 blocks of 6 × 6 plants spaced 30 cm apart (Supplementary Fig. 4). Each block contained four rows of four plants per genotype in north–south (N–S) orientation, surrounded by one border row of WT. WT was present in all blocks ($n = 12$), whereas the four PSBS overexpression and two psbs knock-down lines were randomly assigned to six blocks ($n = 6$). The blocks were positioned in a 3 (N–S) × 4 (E–W) rectangle with 75 cm spacing between blocks. The entire experiment was surrounded by two border rows of WT.

Light intensity (LI-190R quantum sensor, LI-COR, Lincoln, NE, USA) and air temperature (Model 109 temperature probe, Campbell Scientific, USA) were measured nearby on the same field site and half-hourly averages were logged using a datalogger (CR1000, Campbell Scientific, USA). Precipitation was measured at two locations close to the field using precipitation gauges (NOAH IV Precipitation Gauge, ETI Instrument Systems Inc., Fort Collins, CO, USA) (Supplementary Fig. 7). Watering to restore field capacity was provided daily when needed through parallel drip irrigation lines with emitters every 30 cm (17 mm PC Drip Line #DL077, The Drip Store, Vista, CA, USA) spanning the whole experiment in E–W orientation and spaced 30 cm apart in N–S direction. To improve soil drainage after watering and precipitation events, two trenches with a depth of approximately 10 cm were dug in N–S direction between the blocks and connected on the south side of the experiment to a 15 cm deep E–W trench. Photosynthesis measurements were performed on the youngest fully expanded leaf 22 days after transplanting. Plants were harvested on July 7, 2016. At final harvest, stem length and the number of leaves were determined, and leaf area was measured with a conveyor-belt scanner (LI-3100C Area meter, LI-COR, Lincoln NE, USA). Leaf, stem, and root fractions were dried to constant weight at 60 °C in a custom-built drying oven equipped with condenser to further dry the recirculated air, after which the dry weights were determined.

**Non-photochemical quenching in field-grown plants**. Leaf discs were sampled pre-dawn from field-grown plants of psbs-4, psbs-50, PSBS-28, PSBS-34, PSBS-43, PSBS-46, and WT control and stored in darkness in glass vials for up to 4 h until measurement. Humidity in the vials was maintained fully saturated by placing a piece of wet filter paper in each vial. Dark-adapted leaf discs were positioned on a piece of wet filter paper in a chlorophyll fluorescence imager (CFimager, Technologica, Colchester, UK) to determine maximal fluorescence ($F_m$). Subsequently, leaf discs were exposed to 15 min of 1000 µmol m$^{-2}$ s$^{-1}$, after which maximal fluorescence without dark adaptation was determined ($F_m$'). NPQ was then determined according to Eq. 3.

**Photosynthetic gas exchange in field**. The response of photosynthetic gas exchange to light intensity was measured on the youngest fully expanded leaf of four plants of psbs-4, PSBS-28, PSBS-34, PSBS-43, and WT control in the S–W blocks. Measurements were performed in four complete sets to account for random effects of N–S position of plants, and time of day. Leaves were clamped in the cuvette of an open gas exchange system (LI6400XT, LI-COR, Lincoln, NE, USA) and allowed to reach steady-state gas exchange at saturating light intensity of 2000 µmol m$^{-2}$ s$^{-1}$, with block temperature set to 30 °C and [$CO_2$] in the airstream controlled to 400 µmol mol$^{-1}$ and vapor pressure deficit between air and leaf kept below 1.5 kPa. Subsequently, light intensity was varied from 2,000 to 1,500, 1,000, 800, 600, 400, 300, 200, 170, 140, 110, 80, and 50 µmol m$^2$ s$^{-1}$. Due to the limited window suitable for measuring gas exchange in field trials, waiting time for steady state was kept between 5 and 10 min for these measurements. When steady state was reached, net assimilation rate, stomatal conductance, and intercellular [$CO_2$] were logged. After gas exchange measurements were performed, leaf absorptance was determined using an integrating sphere (LI1800, LI-COR, Lincoln, NE, USA) connected to a spectrometer (USB-2000, Ocean Optics Inc., Dunedin, FL, USA).

**Statistical analysis**. All statistical analysis was performed with SAS (version 9.3, SAS Institute Inc., Cary, NC, USA). Data were tested with the Brown–Forsythe test for homogeneity of variance and the Shapiro–Wilk test for normality. One-way analysis of variance was applied to transcription levels, protein expression, gas exchange data, Rubisco content and activation state, stomatal density, and dimension data. Measurement set was included as a random effect in analysis of the

field gas exchange data to account for variation caused by N–S plant position and time of day. Biomass, leaf area, and plant height data were analyzed with a linear mixed model accounting for block and genotype effects with Welch–Satterthwaite adjustment of degrees of freedom to account for the different replication rate of WT (PROC MIXED). Significant genotype effects in ANOVA ($\alpha = 0.05$) were followed by testing of genotype means against WT control ($\alpha = 0.05$), using Dunnett's multiple comparison correction. Correlations between $Q_A$ redox state with $g_s$ and protein levels and with $A_n/g_s$ were evaluated using Pearson's correlation coefficient.

**Data availability**. All relevant data and plant materials are available from the authors upon request. Raw data corresponding to the figures and results described in this manuscript have been deposited at: https://data.mendeley.com/datasets/nsbjps9rkg/draft?a=10508d31-685a-4a62-8f96-cb591c569e97.

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

## Acknowledgements

We thank David Drag and Ben Harbaugh for plant management in greenhouse and field studies, Madeline Steiner and Sydney Gillespie for general assistance during lab work and fieldwork. This research was supported by the Bill and Melinda Gates Foundation [OPP1060461] titled "RIPE—Realizing Increased Photosynthetic Efficiency for Sustainable Increases in Crop Yield". K.K.N. is an investigator of the Howard Hughes Medical Institute and the Gordon and Betty Moore Foundation (through Grant GBMF3070). A. D.B.L. was supported by the National Science Foundation Directorate of Biological Sciences (through Grant PGR–1238030).

## Author contributions

K.G., J.K., K.K.N., and S.P.L. designed experiments. L.L. prepared plasmid. K.G. and J.K. generated transgenic tobacco lines. K.G., K.K., and J.K. performed experiments. A.P.C. and D.R.O. measured Rubisco content and activation state. J.X. and A.D.B.L. measured stomatal density and dimensions of stomatal complex. K.G. and J.K. analyzed experiments. K.G., J.K., K.K., L.L., K.K.N., and S.P.L. wrote the manuscript.

## Additional information

**Competing interests:** The authors declare no competing financial interests.

