## [Peer Review File · Nature Communications]

Reviewers' comments:

Reviewer #1 (Remarks to the Author):

This is a fascinating paper in several respects.

1. It shows that stomatal conductance can be reduced, so conserving water, without compromising photosynthetic carbon assimilation. This is a very important result with implications for crop improvement.
2. It provides support for the hypothesis that the response of stomatal conductance to light intensity is mediated by the redox state of the acceptor side of photosystem II.

The paper is nicely written, the data presented is clear and the discussion of the results obtained put in the correct context. It is a major advance. It is clearly good enough to be published in Nature Communications.

My criticisms and suggestions are minor ones - I would have liked to see more discussion of the findings.

1. It is curious that a linear relationship is found between the estimate of the redox state of QA and the effect, since the fluorescence parameter used is not directly proportional to redox state. Whilst perhaps outside the scope of this paper, it would be useful to learn how the authors think this redox control works? Is it via the redox state of the PQ pool, as in state transitions and involving protein kinase.
2. Do the authors know if the regulation of stomatal conduction is proceeding via redox control within the guard cells, or do they imagine a systemic signal?
3. Given the large effect they observe, are there any difference in the acclimation of the leaves to the growth light intensity? Are there difference in stomatal number for example? This could have important consequences for their interpretation.
4. It is remarkable that the rate of electron transport and carboxylation capacity can compensate for reduced reduced stomatal conductance. How does that happen? Are the rubisco activation state or the delta pH altered?
4. Finally, it would be worth commenting as to why such a simple manipulation apparently has such a beneficial phenotype. NPQ capacity and associated PsbS content is highly variable between species, and hence is selected for. Why do Plants not have a higher content? Why is the penalty of having increased NPQ?

Reviewer #2 (Remarks to the Author):

This is an interesting MS that explores the relationship between redox state of Qa and g_s for improved water use efficiency in the field. Transgenic tobacco with varying quantities of the non-photochemical quenching protein PsBs correlate with g_s , resulting in different plant water use efficiencies. PsbS influences the redox state of the Qa, which has been hypothesised to be a signal for stomatal behaviour and important in the co-ordination between g_s and assimilation rate. The MS is well written and data analyses presented well.

Although the data are mostly correlative in nature a convincing argument for the role of PsbS altering initial stomatal conductance and impacting on water use efficiency is presented. The hypothesis that stomatal behaviour is linked to mesophyll photosynthesis, including the redox state of the plastoquinone pool is not new, therefore further additional background information and findings from previous research in this area such be included in the introduction.

Additionally there are many studies that have examined g_s responses in transgenic plants with alterations in photosynthesis – both Calvin cycle and electron transport – some of the findings from this previous research may conflict with the findings presented here – again these should be acknowledged/discussed within the paper.

To inform readers less familiar with this work it would be useful to include a little more specific information on the role of PsbS in NPQ and electron transport and how it functions to alter Qa redox state.

Were the number or size of stomata different in the transgenic compared with WT?

It appears that g_s was always greater in the under-expressers than the WT and lower at all light levels in the plants overexpressing PsbS.

Were there any differences in cuticular conductance or night-time stomatal conductance that could impact on plant water status?

It is not clear how long the plant/leaf was left to reach steady state at each light level. G_s often needs much longer than assimilation rate to reach a true steady state. The authors discuss early stomatal responses linked to Qa redox status and implications for water use efficiency - what would happen to steady state g_s ?

Can the authors provide an explanation as to why the decrease in g_s influenced growth under the field irrigation conditions when no effect of g_s on A was observed? The fact that the increased WUE was not advantageous is understandable but the reasons behind the detrimental effect on growth are unclear, given that the changes in WUE are driven only by g_s with no changes in A-

this needs to be explained. If there was an increase in NPQ would this divert energy away from photosynthetic processes where it was needed?

Reviewer #3 (Remarks to the Author):

This paper claims that the redox state of the primary electron acceptor of PSII, QA, is a sensing mechanism for stomatal aperture and that modifying this by changing levels of PsBS, a key protein involved in non-photochemical energy dissipation, in transgenic plants, water use efficiency is improved.

There are some major issues with the conclusions from this work which I set out below.

1. The decoupling of Gs from A in transgenic tobacco with altered electron transport or carbon metabolism is not new. Baroli et al 2008 (I could not find this paper in the references list) examined tobacco with reduced Cytochrome B6F level and hence reduced assimilation (and also reduced Rubisco lines which behave similarly) and showed that conductance no longer followed assimilation rate; ie, when ETR and assimilation were very low, Ci remained high and Gs high. These plants had greatly perturbed QA redox states, PhiPSII and ETR yet Gs was unaffected. This seems completely at odds with the current observations.
2. Interpretation of the results in this paper are complicated by the use of a constitutive promoter. If the authors are correct and redox state of QA, presumably controls the red light response of guard cells, is it sensed in the guard cell or in the mesophyll cells? Comparing these results with a guard cell specific gene suppression and overexpression construct might unravel more of the mechanism.
3. The "operating" gs for a given CO₂ concentration and assimilation rate can be "moved" around by altering photosynthetic capacity (transpiration efficiency is a product of both Gs and capacity). While estimates of V_{cmax} and J are given, to appreciate the details of the significance of these observations, full A vs Ci curves should be provided in addition to the light response curves shown.
4. Is stomatal density changed in these lines? I believe that stomatal density changes have been reported in the literature for such transgenic plants with altered ETR and calvin cycle components.
5. While QA redox state appears to correlate well with gs, this is only a correlation and does not establish cause and effect.
6. Rather than a field trial, some lysimetry would have been useful to determine whether the improved TE or instantaneous WUE translated to whole plant water use.

Responses to Reviewers' Comments:

The blue text in the revised manuscript indicates changes/additions that were made to the text.

Response to reviewers' comments:

Reviewer #1 (Remarks to the Author):

1. It is curious that a linear relationship is found between the estimate of the redox state of Q_A and the effect, since the fluorescence parameter used is not directly proportional to redox state.

Answer: To estimate the redox state of quinone A, we used the q_L fluorescence parameter as defined by Kramer *et al.* 2004 to estimate the fraction of Q_A in its oxidized state. The derivation of this parameter assumes a lake model for photosynthetic antenna complexes (i.e., antennae are shared between reaction centers), and represents a modification of the perhaps more familiar q_P parameter which is based on the puddle model (i.e., assuming each reaction center has its own designated antenna complex). The same paper (Kramer *et al.* 2004) and several others have made a convincing case that the lake assumption is more appropriate than the puddle assumption for higher plant thylakoid membranes, although an intermediate situation between both extremes may be most accurate. We have now explained this in more detail in the materials and methods section (l.278-281).

Whilst perhaps outside the scope of this paper, it would be useful to learn how the authors think this redox control works? Is it via the redox state of the PQ pool, as in state transitions and involving protein kinase.

Answer: The presented research was aiming to verify the hypothetical link between Q_A redox and stomatal conductance, and test if this could be used to improve WUE_i under field conditions. As such, as the reviewer indicates, this question is outside the scope of this paper and part of ongoing further research. However, the fact that we observed a clear stomatal conductance phenotype as a result of targeted genetic modulation of Q_A redox via NPQ confirms its role as an early signal for the stomatal light response, and therefore our results facilitate further exploration of the putative signaling pathway in a targeted way that was not previously possible.

2. Do the authors know if the regulation of stomatal conduction is proceeding via redox control within the guard cells, or do they imagine a systemic signal?

Answer: This is an interesting comment that was also mentioned by the other two reviewers (point 1 reviewer 2, point 2 reviewer 3). Although the constitutive (35S) promoter used to drive *PsbS* expression leads to expression in both mesophyll and guard cells and it is likely that Q_A redox state was affected in chloroplasts in both cell types, the fact that the measured chlorophyll fluorescence signal almost entirely originates from the mesophyll parenchyma implicates a mesophyll-derived signal, which would also be consistent with several other studies indicating a link between mesophyll and stomatal opening (reviewed by Lawson *et al.* 2014). However, it cannot be ruled out that guard cells were also more directly involved. We have now included these considerations in the revised manuscript (l.46-50 and l.189-193).

3. Given the large effect they observe, are there any difference in the acclimation of the leaves to the growth light intensity? Are there difference in stomatal number for example? This could have important consequences for their interpretation.

Answer: This is an interesting question that was also raised by the other reviewers (point 3 reviewer 2, point 4 reviewer 3). To address this question, we have done a new experiment to find out if the

overexpression or silencing of PsbS affected stomatal density or dimensions of the stomatal pore complex in the same transgenic and WT lines. No significant differences were found in the length or width of the stomatal pore complex in plants from PsbS overexpressing lines PSBS-28 and PSBS-43, PsbS knock-down lines psbs-4, and tobacco wild type. Stomatal density in PSBS-43 was similar to wild-type and was somewhat reduced in PSBS-28 and psbs-4, relative to wild type. Neither density nor stomatal size measures could explain the PsbS-associated stomatal conductance phenotype that we reported. Therefore, we suggest that our new analysis shows that our findings are most likely explained by a difference in regulation of stomatal conductance, rather than an adjustment in stomatal morphology or density. These new experimental results have been included in the revised manuscript as Supplementary Fig. 3 and are commented upon in the revised text (l. l. 118-123 and l.332-349).

4. It is remarkable that the rate of electron transport and carboxylation capacity can compensate for reduced reduced stomatal conductance. How does that happen? Are the rubisco activation state or the delta pH altered?

Answer: To address this comment, we undertook a new experiment to look at Rubisco content and activation state corresponding to light-saturated, steady-state photosynthesis at ambient CO₂. Rubisco content was not affected by increased or reduced PsbS content. However, a weak positive trend ($P = 0.06$) between PsbS content and Rubisco activation state was found. Activation state of Rubisco in psbs-4 (85%) was lower than in WT (95%), whereas activation states in PSBS-28 and PSBS-43 were slightly higher. These findings are consistent with the weak positive trends observed for V_{cmax} ($P = 0.29$) and J_{max} ($P = 0.07$) with PsbS content and suggest that the compensation in net assimilation rate may be explained to some extent by more active Rubisco. These new experimental results have been included in the revised manuscript as Supplementary Fig. 2 and commented upon in the revised text (l. 105-111, l.205-209 and l.305-331).

5 Finally, it would be worth commenting as to why such a simple manipulation apparently has such a beneficial phenotype. NPQ capacity and associated PsbS content is highly variable between species, and hence is selected for. Why do Plants not have a higher content? Why is the penalty of having increased NPQ?

Answer: Although the improvement in water use efficiency would be beneficial under various conditions, we also show that these benefits are representing a trade-off and may come at a cost with regards to productivity under well-watered field conditions (Fig. 3e-g). Additionally, although no significant differences were found for steady-state CO₂ assimilation rates, it is possible that increasing PsbS may reduce the efficiency of photosynthesis under fluctuating light conditions, which we recently showed to have an appreciable effect on productivity under similar field conditions (Kromdijk *et al.*, 2016). This has been added to the revised manuscript (l.197-201).

Reviewer #2 (Remarks to the Author):

1. Although the data are mostly correlative in nature a convincing argument for the role of PsbS altering initial stomatal conductance and impacting on water use efficiency is presented. The hypothesis that stomatal behaviour is linked to mesophyll photosynthesis, including the redox state of the plastoquinone pool

is not new, therefore further additional background information and findings from previous research in this area such be included in the introduction. Additionally there are many studies that have examined gs responses in transgenic plants with alterations in photosynthesis (both Calvin cycle and electron transport) some of the findings from this previous research may conflict with the findings presented here - again these should be acknowledged/discussed within the paper.

Answer: We agree that we have been rather sparse with references and review of prior work in the previous version of the manuscript (primarily due to the fact that the initial submission was aimed at a shorter format). To address the points raised by the reviewer, we have substantially revised the introduction (l.43-84) as well as the discussion (l.174-193) and provided a new Fig. 4, which schematically summarizes how our results complement prior work in this area and why we think our manipulation via NPQ is specifically suitable for studying the stomatal light response. While a link to photosynthesis has been inferred in many previous studies, this the first to show manipulation of a specific gene that can improve leaf water use efficiency without compromising CO₂ assimilation.

2. To inform readers less familiar with this work it would be useful to include a little more specific information on the role of PsbS in NPQ and electron transport and how it functions to alter Q_A redox state.

Answer: We agree with this suggestion and have revised the introduction to better explain the role of PsbS in NPQ and how it affects electron transport and Q_A redox (l.72-84). We have also prepared a schematic to show how NPQ interacts with the other main electron transport components as well as Calvin-Benson cycle and implemented this as a new Fig. 4.

3. Were the number or size of stomata different in the transgenic compared with WT? It appears that gs was always greater in the under-expressers than the WT and lower at all light levels in the plants overexpressing PsbS.

Answer: See also point 3 from reviewer 1. We performed a new experiment with optical tomography on abaxial and adaxial leaf surfaces of our transgenic lines to estimate stomatal density and pore size. The results show no differences in length, width or length × width of the stomatal pore complex. Slight reductions in stomatal density were found for silencing line psbs-4 and overexpression line PSBS-28 relative to tobacco wild type. These results do not explain the patterns in stomatal conductance we observed during the light response curves, reinforcing our conclusion that the manipulation of stomatal conductance via PsbS acts via the regulation of stomatal opening in response to light intensity. These new experimental results have been included in the revised manuscript as Supplementary Fig. 3 and in the revised text (l. l.118-123 and l.332-349).

4. Were there any differences in cuticular conductance or night-time stomatal conductance that could impact on plant water status?

Answer: We have reanalyzed the dark-adapted stomatal conductance prior to the light response measurements under controlled conditions. Plants were kept in very weak light (<10 μmol m⁻² s⁻¹) prior to measurement and after clamping on the gas exchange cuvette we dark-adapted leaves inside the gas exchange cuvette for 30 minutes before starting light curves. These dark-adapted conductance values thus represent some residual stomatal conductance + cuticular conductance in darkness just prior to the

start of each light response curve. This observed dark conductance varied between 0.11-0.12 mol H₂O m⁻² s⁻¹ and did not differ between lines (See Figure below; $P = 0.89$).

5. It is not clear how long the plant/leaf was left to reach steady state at each light level. g_s often needs much longer than assimilation rate to reach a true steady state. The authors discuss early stomatal responses linked to Q_a redox status and implications for water use efficiency - what would happen to steady state g_s ?

Answer: We agree that g_s can take a long time to stabilize, possibly longer than we may have captured with our measurement protocol. We found that starting at dark-adapted conditions and slowly moving up in light intensity in reasonably small steps gave rise to 'pseudo steady-state' conductance faster than starting at high light and stepwise reducing light intensity, possibly because of the interplay between light and C_i effects at high PFD or large step-changes. In the light response curves under controlled conditions we settled on a waiting time between 10 – 20 minutes per step (determined by stability settings), which led to an overall program length of approximately 3-4 h per curve. This seemed to work well for light-response curves moving from low to high PFD, as evidenced by relatively little change in conductance between two subsequent measurements at the same light intensity (separated by 5-10 minutes), which typically varied less than 5%. The light response curves under field conditions had to be done much faster and thus less stringent stability settings had to be used to fit the measurements inside the small window of time during the day that can be utilized for unstressed gas exchange measurements in a field experiment. Therefore we chose to wait approximately 5-10 minutes per step in these curves. We agree that the response in g_s measured in these curves probably does not reflect the full extent of stomatal movement. In fact g_s often also includes long-term diurnal movements, which likely prevents a 'true' steady state from ever occurring (e.g. Vialet-Chabrand *et al.* 2013 *Plant Cell Env* 36: 1529-1546 and Matthews *et al.* 2017 *Plant Phys* 174: 614-623). In this sense, we clearly measured the short-term component of stomatal response to light intensity, which we have now described more clearly in the materials and methods (l.278-281 and l.295-297). In terms of relevance, light intensity in a field crop canopy is highly dynamic in the time range of seconds to minutes and the response of stomatal conductance to light signals in this time range has clearly been captured to a large extent in our experiments.

6. Can the authors provide an explanation as to why the decrease in g_s influenced growth under the field irrigation conditions when no effect of g_s on A was observed? The fact that the increased WUE was not advantageous is understandable but the reasons behind the detrimental effect on growth are unclear, given

that the changes in WUE are driven only by g_s with no changes in A - this needs to be explained. If there was an increase in NPQ would this divert energy away from photosynthetic processes where it was needed?

(See also point 5 of reviewer 1). It is difficult to conclusively answer this question. We suspect that it wasn't so much the decrease in g_s , but the increase in PsbS and the increased levels of NPQ associated with PsbS overexpression which may have had slight adverse effects on the efficiency of CO₂ assimilation under fluctuating light conditions, as previously shown by Hubbart *et al.* (2012) for rice overexpressing PsbS. We have recently shown that this efficiency has significant implications for biomass productivity of tobacco in similar field conditions (Kromdijk *et al.* 2016). We have now explained this more clearly in the revised manuscript (l.194-201).

Reviewer #3 (Remarks to the Author):

*1. The decoupling of G_s from A in transgenic tobacco with altered electron transport or carbon metabolism is not new. Baroli *et al.* 2008 (I could not find this paper in the references list) examined tobacco with reduced Cytochrome B6/f level and hence reduced assimilation (and also reduced Rubisco lines which behave similarly) and showed that conductance no longer followed assimilation rate; ie, when ETR and assimilation were very low, C_i remained high and G_s high. These plants had greatly perturbed Q_A redox states, PhiPSII and ETR yet G_s was unaffected. This seems completely at odds with the current observations.*

Answer: We have made substantial revisions to the manuscript and now provide a more concise summary of previous literature, including the paper by Baroli *et al.* (2008). We don't agree that our results contradict these observations and have described how we think our results fit in with prior work. Taking the specific case of the results by Baroli *et al.* as an example, as the reviewer mentions, the decrease in Cytochrome b_6/f had a detrimental effect on electron transport and assimilation, leading to strongly decreased CO₂ uptake at a given light intensity. Whereas Q_A redox state is not reported in Baroli *et al.*, this would have likely led to substantially more reduced Q_A (although NPQ would have presumably increased in parallel, which may have buffered this effect by decreasing excitation pressure). According to our findings, more reduced Q_A should have led to increased stomatal opening relative to the wild-type plants, however due to the strong reduction in A_n , C_i was also significantly higher in these plants providing a strong opposing signal for stomatal conductance to decrease. Therefore, it seems that the opposing effects of more reduced Q_A and increased C_i on stomatal conductance in these plants may have resulted in the observed stomatal conductance levels, which were similar to wild type. The same reasoning would also explain why several other genetic manipulations have led to ambiguous results with regards to g_s . We have summarized these in a new Fig. 4 and have revised the manuscript to explain how manipulation of NPQ allows a less confounded view on the link between Q_A redox and stomatal conductance in response to light intensity, in particular compared to mutants impaired in electron transport downstream of Q_A or with impaired Calvin-Benson cycle function, where Q_A redox signals on stomatal conductance are strongly confounded by opposing C_i signals due to impaired CO₂ assimilation (l.54-65 and l.174-193). Our study also differs from these prior studies in identifying manipulation of a specific gene that improves leaf water use efficiency without compromising CO₂ assimilation in any major way, giving it key practical application.

2. Interpretation of the results in this paper are complicated by the use of a constitutive promoter. If the authors are correct and redox state of Q_A , presumably controls the red light response of guard cells, is it

sensed in the guard cell or in the mesophyll cells? Comparing these results with a guard cell specific gene suppression and overexpression construct might unravel more of the mechanism.

Answer: See also point 2 of reviewer 1. We agree that it is hard to specifically state where the signal originates. However, considering the fact that the fluorescence signal largely originates from the mesophyll, the redox state of Q_A in the guard cells would have to correlate linearly with redox state of Q_A in the mesophyll cell chloroplasts to maintain the linear response, which does not seem very likely. Therefore, we would speculate that the signal is probably sensed in the mesophyll, which would be consistent with the large body of work implying the need for a mesophyll-driven signal in the stomatal red light response. We have now included these considerations in the revised manuscript (l.46-50 and l.189-193).

3. The "operating" g_s for a given CO_2 concentration and assimilation rate can be "moved" around by altering photosynthetic capacity (transpiration efficiency is a product of both G_s and capacity). While estimates of V_{cmax} and J are given, to appreciate the details of the significance of these observations, full A vs C_i curves should be provided in addition to the light response curves shown.

Answer: The individual CO_2 response curves were already provided in the raw data repository and have now also been used to produce a new Supplementary Fig. 2a. These curves were used to estimate V_{cmax} , whereas J_{max} (not J) was estimated from the light response curves of whole-chain electron transport, which have now been plotted in Supplemental Fig. 2b. In both cases, weak positive trends were found with the presence of PsbS. Our new analysis of Rubisco activation state and content in these plants suggests that these trends can at least partly be explained by differences in Rubisco activation state. (See also point 4, reviewer 1). These new experimental results have been included in the revised manuscript as Supplementary Fig. 2 and in the revised text (l. 105-111, l.205-207 and l.305-331).

4. Is stomatal density changed in these lines? I believe that stomatal density changes have been reported in the literature for such transgenic plants with altered ETR and calvin cycle components.

This question was also brought up by the other reviewers (point 3 reviewer 1 and point 3 reviewer 2) and prompted us to do a new experiment. Using optical tomography we found that stomatal density was slightly decreased in PsbS silencing line psbs-4, as well as PsbS-overexpressing line PSBS-28, whereas PSBS-43 was unaltered relative to wild-type tobacco. Stomatal length and width and their product were all unaffected by the differences in PsbS content. These results have now been included in the revised manuscript as Supplementary Fig. 3 and in the revised text (l. l.118-123 and l.332-349).

5. While Q_A redox state appears to correlate well with g_s , this is only a correlation and does not establish cause and effect.

Answer: This would be true if we had only evaluated this relationship in wild-type tobacco plants. However, here we specifically manipulated NPQ via PsbS overexpression and silencing, which directly affected Q_A redox state. We would argue that the fact that these changes in Q_A redox state resulted in stomatal conductance responses in the hypothesized direction clearly goes further than mere correlations and instead seem to imply cause and effect, although several components of the signaling pathway are perhaps yet to be discovered.

6. Rather than a field trial, some lysimetry would have been useful to determine whether the improved TE or instantaneous WUE translated to whole plant water use.

Our objective here was to determine if stomatal conductance would be decreased and leaf level instantaneous water use efficiency (iWUE) be improved by manipulation of PsbS expression under field conditions. This is clearly demonstrated in the measurements that we provide. Showing an impact on whole crop water use efficiency by lysimetry, as the reviewer would be aware, would require a major field engineering operation, i.e., to insert weighing lysimeters, which would be a major multi-year project. Insertion of weighing lysimeters causes considerable soil disturbance, and is now recognized to impose significant artifacts, which could make such experiments inconclusive. While it is well-known, due to biophysical feedbacks on leaf temperature and biological feedbacks on leaf area that a decrease in stomatal conductance and increase in iWUE, may exceed the proportional change at the crop level, that there will be an improvement in whole crop water use is clear. In our previous work with soybean crops we used open-air field elevation of CO₂ to artificially lower stomatal conductance and increase CO₂ assimilation, so increasing iWUE – in effect mimicking what we have now achieved genetically. In this work we showed that decreasing leaf level transpiration by 10% averaged over the growing season resulted in a highly significant 8.6% decrease in crop ecosystem evapotranspiration as determined by season-long Bowen-Ratio micro-meteorological measurement of our replicated treatment plots (Bernacchi CJ, Kimball BA, Quarles DR, Long SP, Ort DR 2007 Decreases in Stomatal Conductance of Soybean under Open-Air Elevation of [CO₂] Are Closely Coupled with Decreases in Ecosystem Evapotranspiration. *Plant Physiology*, 143, 134–144). Therefore the answer to the reviewer's question is already known; if you can decrease water use at the leaf level a decrease in crop ecosystem evapotranspiration follows albeit of somewhat smaller magnitude. This has been explained in the revised manuscript (l.167-173).

Reviewers' comments:

Reviewer #1 (Remarks to the Author):

The authors have responded satisfactorily to all my comments and criticisms. I am happy for it to be published without further change.

Reviewer #2 (Remarks to the Author):

The authors have addressed all of my previous comments, greatly improving and strengthening an already very interesting and well written MS. I particularly like the inclusion of the new Fig. 4.

I have a few very minor comments:

Line 57: It would be useful to include the SBPase paper (Lawson et al.) in this list, as this paper specifically examined the influence of reductions in SBPase activity on photosynthesis (and related processes) on stomatal behaviour. This paper also suggested higher g_s values in transgenic plants. Stomatal behaviour was not the main focus of the glyceraldehyde 3-phosphate dehydrogenase paper.

Line 61: "However, the interpretation..." The authors could consider linking this statement to Fig 4, as it is not immediately clear the message the authors are try to convey when you first read the statement.

Reviewer #3 (Remarks to the Author):

Reports of agronomic effects related to photosynthesis in the field are rare and it is important for studies such as the current one to be reported with careful attention to mechanism and reproducibility. The reviewers had remarkably similar comments as to the shortcomings of the study and the authors have paid careful attention to addressing these. While many more questions arise as to the mechanism of this effect on g_s and its utility as an agricultural trait, the current version now addressed the majority of comments raised.

Response to reviewers' comments:

Reviewer #1 (Remarks to the Author):

The authors have responded satisfactorily to all my comments and criticisms. I am happy for it to be published without further change.

Reviewer #2 (Remarks to the Author):

The authors have addressed all of my previous comments, greatly improving and strengthening an already very interesting and well written MS. I particularly like the inclusion of the new Fig. 4. I have a few very minor comments:

Line 57: It would be useful to include the SBPase paper (Lawson et al.) in this list, as this paper specifically examined the influence of reductions in SBPase activity on photosynthesis (and related processes) on stomatal behaviour. This paper also suggested higher g_s values in transgenic plants. Stomatal behaviour was not the main focus of the glyceraldehyde 3-phosphate dehydrogenase paper.

Answer: The citation Lawson et al. 2008 has been added to the Introduction (L63 in the current version of the manuscript).

Line 61: "However, the interpretation..." The authors could consider linking this statement to Fig 4, as it is not immediately clear the message the authors are try to convey when you first read the statement.

Answer: We appreciate the suggestion but prefer to keep Fig. 4 as part of the discussion instead of starting the manuscript with it. Fig. 4 shows graphically how we interpret our findings in the context of previous work. We think this is better suited for the Discussion and not the Introduction section.

Reviewer #3 (Remarks to the Author):

Reports of agronomic effects related to photosynthesis in the field are rare and it is important for studies such as the current one to be reported with careful attention to mechanism and reproducibility. The reviewers had remarkably similar comments as to the shortcomings of the study and the authors have paid careful attention to addressing these. While many more questions arise as to the mechanism of this effect on g_s and its utility as an agricultural trait, the current version now addressed the majority of comments raised.